# Combination of Ipilimumab and Nivolumab in Cancers: From Clinical Practice to Ongoing Clinical Trials

**DOI:** 10.3390/ijms21124427

**Published:** 2020-06-22

**Authors:** Omid Kooshkaki, Afshin Derakhshani, Negar Hosseinkhani, Mitra Torabi, Sahar Safaei, Oronzo Brunetti, Vito Racanelli, Nicola Silvestris, Behzad Baradaran

**Affiliations:** 1Student research committee, Birjand University of Medical Sciences, Birjand 9717853577, Iran; omidkoshki@gmail.com; 2Department of Immunology, Birjand University of Medical Sciences, Birjand 9717853577, Iran; 3Immunology Research Center, Tabriz University of Medical Sciences, Tabriz 5165665811, Iran; afshin.derakhshani94@gmail.com (A.D.); shar.safaee@gmail.com (S.S.); 4Department of Immunology, Faculty of Medicine, Tabriz University of Medical Sciences, Tabriz 5166614766, Iran; hosseinkhanin@tbzmed.ac.ir; 5Student research committee, Tabriz University of medical sciences, Tabriz 5165665811, Iran; mitra.tbm@gmail.com; 6Medical Oncology Unit, IRCCS IstitutoTumori “Giovanni Paolo II” of Bari, 70124 Bari, Italy; dr.oronzo.brunetti@tiscali.it; 7Department of Biomedical Sciences and Human Oncology, University of Bari “Aldo Moro”, 70124 Bari, Italy; vito.racanelli1@uniba.it

**Keywords:** immune checkpoint inhibitors, cancer, nivolumab, ipilimumab, combination therapy

## Abstract

Cytotoxic T-lymphocyte-associated protein 4 (CTLA-4) and programmed cell death protein 1 (PD-1) are inhibitory checkpoints that are commonly seen on activated T cells and have been offered as promising targets for the treatment of cancers. Immune checkpoint inhibitors (ICIs)targeting PD-1, including pembrolizumab and nivolumab, and those targeting its ligand PD-L1, including avelumab, atezolizumab, and durvalumab, and two drugs targeting CTLA-4, including ipilimumab and tremelimumab have been approved for the treatment of several cancers and many others are under investigating in advanced trial phases. ICIs increased antitumor T cells’ responses and showed a key role in reducing the acquired immune system tolerance which is overexpressed by cancer and tumor microenvironment. However, 50% of patients could not benefit from ICIs monotherapy. To overcome this, a combination of ipilimumab and nivolumab is frequently investigated as an approach to improve oncological outcomes. Despite promising results for the combination of ipilimumab and nivolumab, safety concerns slowed down the development of such strategies. Herein, we review data concerning the clinical activity and the adverse events of ipilimumab and nivolumab combination therapy, assessing ongoing clinical trials to identify clinical outlines that may support combination therapy as an effective treatment. To the best of our knowledge, this paper is one of the first studies to evaluate the efficacy and safety of ipilimumab and nivolumab combination therapy in several cancers.

## 1. Introduction

Since the 20thcentury, cancer therapy has been characterized by ups and downs that are not only due to the ineffectiveness of therapies and side effects, but also conditioned by hope and the fact that in many cases, there has been complete remission [1]. Chemotherapy was the first identified therapeutic approach that systemically delivered chemical agents into tumor tissue and destroyed the large mass, but did not eliminate the disease [2]. In addition to short-term improvement in survival, this treatment has some limitations, such as the high risk of toxicity for rapidly renewing cells, like skin and gastrointestinal cells, and blood stem cells [3]. Moreover, several chemotherapies lead to the development of resistance of the cancer cells, thus limiting their use as monotherapy in further treatment lines [4]. Radiation therapy was considered an important part of cancer therapy, with nearly 50% of all cancer patients undergoing radiation therapy during the progression of the disease [5]. In recent years, immunotherapy has metamorphosed into an important therapeutic option and is now the first choice in many cases. However, depression, fatigue, dermatitis, cardiovascular disease, mucositis and esophagitis, and pneumonitis are some of the important side effects of radiation therapy in different cancers [6,7,8]. More than one century ago, the ability of the immune system to act against tumors was discovered by Dr. William Coley in 1893, when he used bacteria as a stimulator for the immune system to treat cancer [9]. Tumor cell growth and progression are related to immune suppression and they can activate several immune checkpoint pathways that have suppressive roles. In particular, cytotoxic T lymphocyte antigen-4 (CTLA-4) and programmed cell death protein 1(PD-1) are two important immune checkpoints that were formerly identified as molecules performing a role in apoptosis, T cell activation, and the maintenance of acquired immune system tolerance [10]. Recently, a wide range of monoclonal antibodies blocking immune checkpoints has appeared as potent agents in the oncological models [11]. Monoclonal antibodies targeting PD-1 (including pembrolizumab and nivolumab) as well as PD-L1 (including avelumab, atezolizumab, and durvalumab,) and those targeting CTLA-4 (including ipilimumab and tremelimumab) [12], have been approved by the FDA for several cancers, such as melanoma, renal cell cancer, lung cancer, and colorectal cancer [13]. Nivolumab, a fully human immunoglobulin G4 anti-PD-1 monoclonal antibody that was created from Chinese hamster ovary cells, is approved for multiple advanced tumors, including melanoma, non-small cell lung cancer (NSCLC), renal cell cancer, Hodgkin’s lymphoma, squamous head and neck cancer, and urothelial carcinoma [14]. Pembrolizumab is an effective, fully humanized immunoglobulin G4 anti-PD-1 antibody used in cancer immunotherapy [15]. Recently, pembrolizumab plus lenvatinib received approval in treating patients with specific types of endometrial carcinoma [16] and bladder cancer [17]. Atezolizumab is a fully humanized immunoglobulin G1 anti-PD-L1 antibody used in cancer immunotherapy [18]. In 2016 and 2017, it was approved by the FDA for urothelial carcinoma [19] and as first-line treatment for advanced bladder cancer, respectively [20]. Avelumab is a fully humanized anti-PD-L1 antibody used in urothelial carcinoma, Merkel cell carcinoma, and renal cell carcinoma [21]. Durvalumab is a human immunoglobulin G1, an anti-PD-L1 monoclonal antibody that is approved for the treatment of advanced bladder cancer and NSCLC [22,23]. Preclinical and clinical studies have begun to investigate immunotherapeutic strategies in combination with chemotherapy and radiation [24]. However, there are many challenges in the application of these molecules. For example, a large proportion of patients (~80%) do not respond to ICI treatment or some of them develop resistance to therapy [25]. Herein, we review data concerning the clinical activity and the adverse events of ipilimumab and nivolumab combination therapy, assessing ongoing clinical trials to identify clinical outlines that may support combination therapy as an effective treatment. This review—by addressing the primary details of CTLA-4 and PD-1 pathways and the results from clinical studies that evaluated the combination of ipilimumab and nivolumab—aims to support future research in combination therapy as the new standard of care for cancer treatment. To the best of our knowledge, this paper is one of the first studies to evaluate the efficacy and safety of ipilimumab and nivolumab combination therapy in several cancers.

## 2. Methodology

We performed literature searches with PubMed and Google Scholar using the keywords and MeSH terms combination therapy, immune checkpoint inhibitors, immune checkpoint agonists, and ipilimumab and nivolumab combination therapy. Since ClinicalTrials.gov is the largest trial registry in the world, we searched ClinicalTrials.gov for ongoing and completed clinical trials until January 2020. We focused on clinical trials of ipilimumab and nivolumab combination therapy in several cancers, including melanoma, advanced renal cell carcinoma, colorectal cancer, breast cancer, lung cancer, esophageal cancer, hepatocellular carcinoma, Hodgkin’s lymphoma, head and neck cancer, and urothelial carcinoma. Exclusion criteria included studies and clinical trials focusing only on ipilimumab and nivolumab, trials focusing on the pediatric population, and non-interventional trials.

## 3. Cytotoxic T Lymphocyte Antigen-4(CTLA-4)

CTLA-4 was recognized by Brunet et al. from mouse T cell-derived cDNA libraries [26]. CTLA-4 is the earliest known immune checkpoint and in vivo experiments showed that mice died after 3–4 weeks *CTLA-4* deletion for immunosuppression, showing its important roles in immune responses and T cell activation [27]. Activated T cells and Foxp3+ T-reg cells led to *CTLA-4* upregulation, with a key role in self-tolerance and maintaining homeostasis. CTLA-4 is a CD28 homolog and with high affinity binding to B7-1/2. CTLA-4 has a barrier function to prevent T cell activation and proliferation [28]. Numerous investigations provided data that CTLA-4 is linked to autoimmune diseases such as Graves’ disease, type 1 diabetes, thyroiditis, and lupus erythematosus. More recently, CTLA-4 blockade has been demonstrated to be a curative strategy for cancer therapy through the challenge with the CD28-B7 combination to exhibit an inhibitory effect on signaling molecules in a variety of cancer diseases [29]. Tremelimumab is another CTLA-4 inhibitor [30]. Tremelimumab is a fully human IgG2 isotype monoclonal antibody used against CTLA-4 and is under investigation as a treatment for several cancers, including melanoma, mesothelioma, and NSCLC [31,32,33]. Recently, monoclonal antibodies against CTLA-4, ipilimumab, and tremelimumab, alone or in combination with PD-1/L-1 inhibitors, significantly increased antitumor effects and improved the survival of several malignancies (Figure 1).

## 4. Ipilimumab Pharmacology

Ipilimumab is a fully humanized monoclonal anti-CTLA-4 antibody that was approved by the FDA in 2011 for the late-stage of melanoma [34]. In earlier surveys, ipilimumab was commonly used as the treatment of malignant melanoma by 60% of patients in the USA and 40% of patients in European countries [35]. In 2017, it was approved for use in pediatric cases with a history of metastatic melanoma. Studies showed a positive effect of ipilimumab when combined with other agents, including vaccines or other immune checkpoint inhibitors against cancer. The FDA approved the positive results of ipilimumab in combination with nivolumab for metastatic melanoma, metastatic colorectal cancer, and advanced renal cell carcinoma [36,37,38]. Hodi FS et al. discovered ipilimumab as a safe and active treatment. All patients in this study had metastatic melanoma that could not be surgically removed [39]. In this study, 676 metastatic melanoma patients were randomly treated with ipilimumab (3 mg/kg) plus gp100 (403 patients), ipilimumab alone (137), or gp100 alone (136). Ipilimumab was administered with or without gp100 every three weeks for up to four treatments. Based on their results, ipilimumab presented a strong response and stable disease (SD) rate in patients who received treatment. The recommended dose of ipilimumab monotherapy for unresectable/metastatic melanoma is 3 mg/kg with intravenous (IV) administration, over 90 min, every three weeks with a maximum of four doses. In addition, the recommended dose of combination therapy for renal cell carcinoma and colorectal cancer is IV administration of 1 mg/kg ipilimumab over 30 min, following nivolumab administered on the same day, every three weeks with up to four doses or until intolerable toxicity or disease progression [40]. Ipilimumab has many side effects, such as fatigue, diarrhea, skin rash, endocrine deficiencies, and colitis. Additionally, 12.9% of patients showed autoimmune reactions [41].

## 5. Programmed Cell Death Protein 1 (PD-1)

The surface receptor PD-1 (CD279) was discovered for the first time in 1992 on a murine T cell hybridoma [42]. *PD-1* is expressed on CD4+ and CD8+ T cells, B cells, monocytes, NK cells, and DCs and leads to inhibition of proliferation, differentiation, and cytokine secretion of T cells [43]. PD-1, which is a family of immunoglobulin domain (Ig) co-receptors, primarily participates in the inhibition of the acquired immune system, especially T cells in several conditions such as malignancy [44]. Previous reports showed a correlation between the expression of *PD-1* and prognosis in cancer patients [45].

## 6. Nivolumab Pharmacology

Nivolumab is a fully humanized monoclonal anti-PD1 antibody that interacts with its ligands PD-L1 and PD-L2 and has an indispensable role in fine-tuning T cell function and maintaining immune system homeostasis. Nivolumab is a genetically engineered monoclonal antibody produced from ovary cells of the Chinese hamster [46]. In 2014, nivolumab received its first Food and Drug Administration (FDA) approval for patients with unresectable or advanced melanoma who did not respond to other therapies [47]. Subsequently, in 2015, the FDA approved the use of nivolumab to treat lung cancer [48]. Until February 2020, nivolumab monotherapy or combination with ipilimumab received FDA approval for use in several different cancers, including NSCLS and small cell lung cancer (SCLS), renal cell carcinoma (RCC), Hodgkin’s lymphoma (HL), head and neck cancer (HNC), urothelial carcinoma (UC), colorectal cancer (CRC), and hepatocellular carcinoma [49,50,51].The usual adult dose for nivolumab is 240 mg IV every two weeks over 30 min until disease progression and 480 mg IV every four weeks over 30 min. Nivolumab affects several body organs including skin, liver, gastrointestinal, respiratory system, endocrine, and cardiovascular systems, and has many common side effects (called immune-related adverse events (irAEs)) [52]. Dermatologic side effects including rash (21%), pruritus (19%), vitiligo (11%), and erythema (10%) [53] also occur. Hepatic side effects (elevated AST (28%), elevated alkaline phosphatase (22%) [54], elevated ALT (16%)) appear along with gastrointestinal side effects (diarrhea or colitis (21%)) [55], respiratory side effects (cough (17%), upper respiratory tract infection (17%)) [56], cardiovascular side effects (peripheral edema (10%)), and endocrine side effects (hypothyroidism and hyperthyroidism) [57] (Figure 2).

The activation of T cells is mediated by the interaction of the TCR and the CD28 receptor with an MHC-II and B7 co-stimulatory molecule located on the APCs. The CTLA-4: B7 binding delivers an inhibitory signal that is effectively inhibited by anti-CTLA-4 antibodies. On the other side, the PD-1: PD-L1 binding between T cells and tumor cells is prevented by anti-PD-1/PD-L1 antibodies. The figure was produced using Servier Medical Art (http://smart.servier.com/). Abbreviations: PD-1, programmed death receptor-1; PD-L1, programmed cell death receptor ligand-1; TCR, T cell receptor; MHC II, major histocompatibility complex class II.

## 7. Combination Therapy in Different Cancers

### 7.1. Combination Therapy in Melanoma

Melanoma staging is based on tumor width, ulceration, lymph node, and metastases. Interleukin (IL)-2 treatment was the first approved immunotherapy for use in melanoma, prescribed alone, or combined with other drugs. Progress in immunotherapy and the development of T cell ICIs such as CTLA-4 and PD-1 inhibitors have further limited the use of IL-2 as a treatment for melanoma patients in high doses [58]. The positive effect of using ipilimumab alone or in combination with nivolumab in melanoma patients with metastasis has been shown in several clinical trials. Larkin J et al. investigated the combination of ipilimumab and nivolumab or single therapy in untreated melanoma patients. The authors randomly enrolled more than 940 untreated melanoma patients in stages III and IV, dividing them into three groups according to the type of treatment used: nivolumab alone (3 mg/kg every two weeks); nivolumab (1 mg/kg) plus ipilimumab (3 mg/kg) every three weeks with four doses followed by nivolumab (3 mg/kg every two weeks);ipilimumab alone (3 mg/kg every three weeks with four doses). Median progression-free survival (mPFS) was 11.5 months for nivolumab plus ipilimumab, 2.9 months for ipilimumab alone, and 6.9 months for nivolumab alone. In addition, in PD-L1-positive patients, mPFS was 14.0 months in both the nivolumab plus ipilimumab and nivolumab alone groups, In PD-L1-negative patients, however, mPFS was longer with the combination as compared with nivolumab alone (11.2 months vs. 5.3 months) [38]. Another phase 2 trial randomly assigned patients with brain melanoma in groups as follows: nivolumab (1 mg/kg) plus ipilimumab (3 mg/kg) for three weeks, followed by nivolumab (3 mg/kg) for two weeks. Their results showed 26% complete response (CR), 30% partial response (PR), and the rate of SD was 2% for six months. Nivolumab combined with ipilimumab had clinically significant intracranial efficiency [59]. In addition, Kevin Diao et al. investigated the role of ipilimumab and stereotactic radiosurgery in melanoma brain metastases. Of the 91 melanoma patients enrolled in their study, 33 patients were treated with ipilimumab in combination with radiosurgery, 28 patients non-concurrently, and 40 patients without ipilimumab. Concurrent ipilimumab administration was defined as within ± 4 weeks of the radiosurgery procedure. The median follow-up time was 7.4 months. Their results showed that OS was increased by 15.1 months in patients who received ipilimumab as compared to 7.8 months in other patients [60]. Their findings confirm that the positive effect of ipilimumab and radiosurgery can be considered clinically suitable. Table 1 summarized some of the completed (until February 2020) clinical trials of ipilimumab plus nivolumab in melanoma.

### 7.2. Combination Therapy in Advanced Renal Cell Carcinoma

Clear renal cell carcinoma (RCC) is a common type of kidney cancer. Approximately90% of all kidney cancers evolved into renal cell carcinomas [66]. For years, VEGFR tyrosine kinase inhibitors (TKIs)were the main treatments for RCC [67]. Recently, ipilimumab received FDA approval in renal cell carcinoma therapy. The combination of ipilimumab and nivolumab was first investigated in a phase II study in patients with metastatic RCC, on various dosing plans [68]. Yang JC et al. in a phase II clinical trial, investigated the role of ipilimumab (3 mg/kg, three weeks) in two cohorts with RCC. Two cohorts received either 3 mg/kg followed by 1 mg/kg or all doses at 3 mg/kg every three weeks. The ORR was 5% in cohort A (3 mg/kg loading dose followed by 1 mg/kg every three weeks) and 12.5% in cohort B (3 mg/kg every three weeks for all doses) [69]. Furthermore, in a phase 3 CheckMate214 trial, Motzer RJ et al. found that the combination of ipilimumab plus nivolumab has a higher OS (75% vs. 60%) and a higher response to the therapy than those people who received sunitinib alone. They enrolled a total of 1096 patients to receive either nivolumab (3 mg/kg) plus ipilimumab (1 mg/kg) intravenously every three weeks with four doses, followed by nivolumab (3 mg/kg) every two weeks, or sunitinib (50 mg) orally once daily for four weeks (six-week cycle) [70]. Some studies have reported adverse, non-threatening events including hypophysis related to ipilimumab [71]. On the whole, the emerging data confirm the clinical efficiency of combination immunotherapy in RCC. Further investigations are needed before consideration for market approval. Table 2 summarized some of the ongoing (until February 2020) clinical trials of ipilimumab plus nivolumab in RCC.

### 7.3. Combination Therapy in Colorectal Cancer

Colorectal cancer (CRC)is the third most deadly and fourth most commonly diagnosed cancer in the world. Nearly 2 million new cases and about 1 million deaths were reported in 2018 [72]. Microsatellites (also known as short tandem repeats (STRs) are small (1–6 base pairs), copying stretches of DNA spread throughout the entire genome,) account for nearly 3% of the human genome, and are responsible for high mutation rate. Microsatellite instability (MSI) is the result of a defective DNA mismatch repair (MMR) system and can be found in CRC. About 15% of all CRC cases have MSI and show a response to immunotherapy with monoclonal antibodies [73]. Recently, the FDA approved a combined therapy with ipilimumab plus nivolumab in CRC patients with MSI. CheckMate142, an ongoing phase 2 trial, evaluated a combination therapy of nivolumab plus ipilimumab for MSI-high subtype of CRC. This study enrolled 119 previously treated patients and gave them nivolumab (3 mg/kg) plus ipilimumab (1 mg/kg) once every three weeks followed by nivolumab 3 mg/kg once every two weeks. The primary endpoint was ORR and secondary endpoints were an objective response, disease control, PFS, and OS. Objective responses were recorded in 65 (55%) of 119 patients and 80% of patients (*n* =95) had disease control for 12 weeks or longer. At 12 months, PFS was 71% (95% CI 61.4–78.7) and OS was 85%. In the nivolumab-only cohort, 31% of patients had an objective response and 12-month OS was 73% [74]. Among 119 patients, 25% had an endocrine, 23% had a gastrointestinal, 19% had hepatic, 5% had pulmonary, 5% had renal and 29% experienced skin irAEs; the majority (57%) were grade 1/2 [75]. The result of this study indicated nivolumab plus ipilimumab as a new treatment option for patients with MSI-high CRC. Now, several ongoing trials are investigating a combination therapy of nivolumab plus ipilimumab in CRC (Table 3).

### 7.4. Combination Therapy in Breast Cancer

Breast cancer is the second most common cancer in newly-diagnosed patients, affecting women worldwide, and its frequency and mortality rates are supposed to increase significantly in the future [76]. Genetic, environmental and lifestyle factors such as a high-calorie diet, alcohol consumption, and lack of physical activity are the most common cause of breast cancer [77]. Studies suggested that approximately 90–95% of breast tumors are caused by environmental factors, whereas in other cancers, this rate is 70% [78]. Contrast-enhanced (CE) digital mammography, ultrasonography, magnetic resonance imaging (MRI), positron emission tomography, and self-examination of the breast are important and suitable methods of early detection of breast tumors [79].

Conventionally, breast cancer is not a highly immunogenic cancer when compared with tumors such as melanoma and NSCLC, but detection of tumor-infiltrating lymphocytes (TILs) in triple-negative breast cancer (TNBC) and HER2+ breast tumors, and the association of these lymphocytes with pathological complete response (pCR) to treatment, highlighted the importance of the immune system in breast cancer [80]. Asano Y et al. assessed how the expressions of immune checkpoint proteins affected responses to NAC in breast cancer. A total of 177 patients with early-stage breast cancer were treated with NAC and the expression of *PD-L1*, *PDL-2*, and *PD-1* was evaluated by immunohistochemistry. All patients received NACs, including four courses of FEC100 (fluorouracil, epirubicin, and cyclophosphamide) every three weeks, followed by 12 courses of paclitaxel. According to their results, 20.9% of the patients had high *PD-1* expression, 23.7% of the patients had high *PD-L1* expression, and 29.4% of the patients had high *PD-L2* expression. Higher expressions of *PD-1*/*PD-L1* were associated with a higher rate of TNBC, and a lower rate of disease-free survival in TNBC, suggesting the potential role of *PD-1*/*PD-L1* expressions to predict treatment responses [81]. Recently, immunotherapy with several ICIs has emerged as a hopeful and developing area of interest in breast cancer.

In a phase 2 trial, Voorwerk L et al. assessed the sensitivity of PD-1 blockade with nivolumab in patients with TNBC positive breast tumors. In this study, 67 patients were randomized to receive nivolumab plus irradiation and chemotherapy agents including cyclophosphamide, cisplatin, and doxorubicin. The results showed that treatment with cisplatin and doxorubicin had higher ORR responses in patients (23% and 35%, respectively). This study also showed that treatment with doxorubicin and cisplatin may cause a more positive response to nivolumab in TNBC [82]. Several ongoing trials are considering a drug combination of nivolumab and ipilimumab as a possible treatment in breast cancer. In NCT03742986, the aim is to assess the efficacy of nivolumab with NAC in patients with inflammatory breast cancer. In this phase II trial, 52 patients will be divided into two groups, HER2-negative and HER2-positive. HER2-negative patients will be treated with nivolumab (360mg, every 21-day cycle), paclitaxel (80mg/m^2^, on day 1, 8, 15 of every 21 day cycle), with doxorubicin (60 mg/m^2^, every 14 day cycle), and cyclophosphamide (600mg/m^2^). HER2-positive patients will be treated with the treatments mentioned plus paclitaxel (80mg/m^2^ on day 1, 8, 15), trastuzumab (8 and 6 mg/kg), and pertuzumab (840 and 420mg). The pCR is the primary result of this study. In another trial, the aim is to study carboplatin plus nivolumab in metastatic TNBC (NCT03414684). The NIMBUS trial is studying a combination of nivolumab and ipilimumab for hypermutated HER2-negative breast cancer. In this phase II trial, 30 participants will receive ipilimumab (IV, every six weeks) and nivolumab (IV, every two weeks). The primary endpoint is the ORR and secondary endpoints are the ORR of the combination according to immune-related response criteria, clinical benefit rate (CBR), PFS, and OS (NCT03789110). The purpose of NCT03546686 is to determine the efficacy of cryoablation, ipilimumab, and nivolumab versus standard care on three-year OS, in patients with hormone receptor-negative, HER2-negative advanced breast cancer after neoadjuvant chemotherapy. In this phase II study, a total of 160 patients will receive ipilimumab 1–5 days before core biopsy and cryoablation, and nivolumab 1–5 days before core biopsy and cryoablation and every two weeks post-surgery. The primary endpoint is the OR and the secondary endpoints are invasive disease-free survival, distant disease-free survival, OS, and overall safety (NCT03546686). Another phase IIb trial is conducted to assess chemotherapy agents (pegylated liposomal doxorubicin and cyclophosphamide) combined with ipilimumab and nivolumab in treating breast cancer. A total of 75 patients will be divided into two arms. Arm A will be treated with pegylated liposomal doxorubicin plus cyclophosphamide and Arm B will be treated with these chemotherapy agents plus ipilimumab and nivolumab. The primary endpoints are toxicity and PFS, and the secondary endpoints are the duration of response, OS, toxicity, ORR, CBR, and *PD-L1* expression (NCT03409198).

### 7.5. Combination Therapy in Lung Cancer

Lung cancer, including two main subtypes of small-cell lung cancer (SCLC) and non-small-cell lung cancer (NSCLC), is one of the leading causes of cancer death worldwide, especially in China [83]. Novel and effective treatment for SCLC patients has not significantly improved survival in these patients. SCLC cells are very sensitive to chemotherapy agents, but over 70% of patients diagnosed with limited disease SCLC (LD-SCLC) and approximately all patients with extensive-disease SCLC (ED-SCLC) finally develop a spreading disease [84]. ICIs may contribute to wider and better applications in SCLC than vaccines and chemotherapy because these antibodies inhibit T cell activity directly [85]. The combination of ipilimumab with chemotherapy was the first investigation regarding immune checkpoint blockers in SCLC. In a phase 1/2 clinical trial, Antonia SJ et al. evaluated the activity of ipilimumab alone and ipilimumab plus nivolumab in SCLC patients. A total of 216 patients were enrolled and treated as follows: 98 patients with nivolumab (3 mg/kg), 64 patients with combination therapy (1 mg/kg, each of them), every three weeks for four cycles, and 54 patients with combination therapy (3 mg/kg nivolumab+ 1 mg/kg ipilimumab). The median follow-up for patients continuing in the study was 198.5 days [86]. In a phase 3 randomized trial, Reck M et al. evaluated the effect of ipilimumab plus etoposide and platinum vs. chemotherapy plus placebo in 1132 SCLC patients; 954 received at least one dose of study therapy (chemotherapy plus ipilimumab, *n* = 478; chemotherapy plus placebo, *n* = 476). OS was 11 months for ipilimumab plus chemotherapy vs. 10.9 months for the control group (*p*-value = 0.37). Based on their results, ipilimumab plus chemical agents did not increase OS when compared to chemotherapy alone in SCLC patients [87]. In the phase 2 trial, Arriola E et al. assessed the safety and positive effects of ipilimumab plus chemotherapy in SCLC patients [88]. A total of 42 patients were enrolled in this study and treated with carboplatin and etoposide for up to six cycles. A dose of10 mg/kg, ipilimumab was given on day 1 of cycles 3 to 6 and every 12 weeks. Median OS was 17 months and the ORR was 84.8%. They found that ipilimumab in combination with chemotherapy agents such as Carboplatin and Etoposide was an advantage for SCLC patients in the advanced stage. The rate of ipilimumab toxicity (69%) was also associated with a high death ratio in this study. In summary, ipilimumab was the first ICI identified as a treatment for SCLC. It shows beneficial effects even though there is significant toxicity. The recommended dose of combination therapy for the metastatic stage of NSCLC is IV administration of nivolumab 3 mg/kg over 30 min every two weeks and ipilimumab 1 mg/kg over 30 min every six weeks. More clinical trials are needed to verify beneficial results. Table 4 shows some of the ongoing clinical trials regarding the effect of combination treatment with ipilimumab and nivolumab in SCLC patients.

NSCLC is the most common malignant lung cancer (84%) worldwide. In 2013, over 33,000 new cases of NSCLC were identified in the USA [89]. NSCLC has three subtypes: adenocarcinoma (the most common form), squamous cell lung cancer (SQCLC; the second common form), and large-cell carcinoma [90]. The standard systemic care for NSCLC patients were platinum-based chemotherapy combinations, with objective responses of between 30 and 60%, but OS is limited (nearly nine months) [91]. In recent years, huge developments have been made in the treatment of NSCLC patients. These treatments include a blockade of CTLA-4 or a combination of ipilimumab with nivolumab and chemotherapy. In the phase II trial, Yi et al. investigated the immunologic outcomes of ipilimumab plus chemotherapy in the early-stage of NSCLC patients [79]. A total of 24 patients received NAC consisting of three cycles of paclitaxel with either cisplatin or carboplatin and ipilimumab included in the last two cycles. Cycle 1 featured paclitaxel (175 mg/m^2^) with either cisplatin (75 mg/m^2^) or carboplatin (AUC = 6, capped at 900 mg) without ipilimumab, and cycles 2 and 3 featured combination chemotherapy (as in cycle 1) plus ipilimumab (10 mg/kg). The results showed that ipilimumab therapy alone had an insignificant effect on the rates of circulating regulatory T cells and myeloid-derived suppressor cells (MDSCs). In the phase III trial, Govindan R et al. investigated the efficiency and safety of treatment with ipilimumab plus chemotherapy in advanced NSCLC. In a total of 749 randomly enrolled patients, 388patients received one dose of chemotherapy plus ipilimumab (Arm A), and 361 patients received chemotherapy plus placebo (Arm B). Median OS was 13.4 months for arm A and 12.4 months for arm B. The mPFS was 5.6 months for both groups [92]. Hellmann MD et al. in a phase 3 trial, assessed the benefit of ipilimumab plus nivolumab in NSCLC patients. A total of 1739 patients were enrolled in this part of CheckMate 227. Of the 1189 patients who had a PD-L1 expression level of 1% or more, 396 received nivolumab (at a dose of 3 mg/kg every two weeks) plus ipilimumab (at a dose of 1 mg/kg every six weeks), 396 received nivolumab monotherapy (240 mg every two weeks), and 397 received chemotherapy every three weeks for up to four cycles. Of the 550 patients with a PD-L1 expression level of less than 1%, 187received nivolumab plus ipilimumab, 177 received nivolumab plus chemotherapy, and 186 received chemotherapy, nivolumab monotherapy, or platinum doublet chemotherapy. The median OS was 17.1 months in the combination group versus 14.9 months in the chemotherapy group, with two-year OS rates of 40.0% and 32.8%, respectively; besides, 32.8% of patients in the combination group experienced grade 3 or 4 adverse events versus 36.0% in the chemotherapy [48].

NSCLC therapy with ipilimumab plus nivolumab combination is currently under investigation in several trials (Table 5).

### 7.6. Combination Therapy in Esophageal Cancer

Esophageal cancer (EC) is one of the common cancers worldwide and is the 6th cause of cancer-related deaths, with OS varying from 15–20% [93]. Today, EC treatment includes mainly surgery, chemotherapy, radiotherapy, and a combination of them. Currently, ICIs are popularly used in many human tumors, including EC. A combination of ipilimumab plus nivolumab has shown a synergistic effect in clinical models of EC. In CheckMate032, Janjigian YY et al. evaluated the safety and effects of nivolumab and nivolumab plus ipilimumab in patients with EC. In this study, of the 169 patients enrolled, 59 patients received nivolumab (3 mg/kg), 49 patients received nivolumab (1 mg/kg) plus ipilimumab (3 mg/kg), and 52 patients received nivolumab (3 mg/kg) plus ipilimumab (1 mg/kg). The ORR was 12%, 24%, and 8% in the three groups, respectively. In addition, 12-month PFS rates were 8%, 17%, and 10%, respectively; 12-month OS rates were 39%, 35%, and 24%, respectively. Collectively, nivolumab/ipilimumab combination therapy showed significant antitumor function, long-lasting responses, and a safety profile in EC patients who do not respond to monotherapy with chemotherapy agents [94].

Currently, ongoing trials are evaluating the combination of ipilimumab and nivolumab in EC (Table 6).

### 7.7. Combination Therapy in Hepatocellular Carcinoma

Hepatocellular carcinoma (HCC) is the most common primary liver cancer and the leading cause of cancer deaths worldwide [95]. Until 2008, there was no systemic treatment to lengthen the survival rate of HCC patients. Recently, the presence of tumor-infiltrating lymphocytes (TILs) expressing PD-1 in HCC tumors suggests that immunotherapy with ICIs might be helpful in the treatment of HCC [96]. In a phase 1/2 trial (CheckMate040), El-Khoueiry AB et al. investigated the efficacy of nivolumab in HCC patients. A total of 262 patients were treated (48 patients in the dose-escalation phase and 214 in the dose-expansion phase); 202 (77%) of 262 patients have completed treatment. Nivolumab 3 mg/kg was adopted for dose expansion. The ORR was 20% in patients treated with nivolumab in the dose-expansion phase and 15% in the dose-escalation phase [97]. The recommended dose of combination therapy for HCC is IV administration of 3 mg/kg ipilimumab over 30 min, following nivolumab on the same day, every three weeks with up to four doses or until severe toxicity or disease progression. Several ongoing trials are investigating nivolumab/ipilimumab combination in HCC (Table 7).

### 7.8. Combination Therapy in Hodgkin’sLymphoma

Hodgkin’s lymphoma (HL) is a type of B cell cancer affecting around 10,000 new cases each year [98]. This disease originates from lymphocytes, which are components of the immune system and includes 11% of all lymphomas diagnosed in the USA [99]. First-line therapy, which includes chemotherapy followed by radiation therapy has been beneficial in HL patients, but a decrease in toxicity of treatments with an improved OS and outcomes are the main challenges of HL therapy [100]. Parameters that play a significant role in identifying suitable treatment include (1) Establishing the histological characteristics of the disease (classical HL relative is more suitable than nodular lymphocyte-predominant HL); (2) The stage of cancer, in particular, if the patient has a primary or developed-stage disease; (3) The existence of diagnostic indicators that suggest a low survival rate; (4) The occurrence or absence of clinical signs [101]. Currently, ICIs and other immunotherapies are treatments that are being assessed in several trials because of their potency against lymphoid malignancies, especially for cases that do not respond to high-dose therapies and those in which stem cell transplantation has failed. Genetic modifications of the *PD-L1*/*PD-L2* along with *JAK2* locus in the 9p24.1 region have been described and could contribute to the overexpression of *PD-L1* and *PD-L2* [102]. Anti-PD1 antibodies demonstrated a better efficacy and safety profile than ipilimumab, and therefore, might be a better choice for clinical targeting of HL [103]. In a recruiting phase I/II trial, the side effects and the most suitable dose of ipilimumab and nivolumab, when in combination with brentuximab vedotin will be investigated. The investigation will include assessing if a nivolumab monotherapy or a combination with ipilimumab will destroy more tumor cells in HL (NCT01896999). In another recruiting phase 1 trial, the aim is to assess the side effects and the most suitable dose of nivolumab when combined with ipilimumab in HIV-associated classical HL. On the first day, patients will receive nivolumab over 30 min. In addition, patients in dose level 2 will receive ipilimumab over 90 min. Cycles will be renewed every 14 days (in 46 cycles). The primary endpoints of this trial are to establish the maximum tolerated dose (MTD) of each combination and CR rate and the secondary endpoints are to establish a partial response (PR), ORR, duration of response, and PFS. Nonetheless, nivolumab as monotherapy in recurrent or refractory cases has shown significant clinical response [104]. Ongoing and randomized trials with novel combinations of ICIs will probably report promising approaches, which can lead to improved cure rates in HL.

### 7.9. Combination Therapy in Head and Neck Cancer

Head and neck cancer (HNC) is a complex condition including several tumors including cancers involving the hypopharynx, oropharynx, lip, oral cavity, nasopharynx, and larynx that are responsible for 1–2 percent of deaths due to cancer [105]. The consumption of alcohol and the use of tobaccocause at least 75 percent of head and neck cancers and account for around 4% of all malignancies in the USA and less than 5% of all cancers worldwide [106]. Since HNC treatment is heterogeneous and includes many versions, an integrative strategy is needed. Nowadays, surgery, radiation therapy, chemotherapy, chemoradiotherapy, and anti-EGFR cetuximab are the main treatment options in HNC therapy [107]. However, they have some limitations. For example, resistance to radiotherapy remains an important cause of poor survival rates in HNC, or EGFR inhibitors have poor efficacy with resistance to the innate and acquired immune systems [108]. Recently, immunotherapy has emerged as a new and promising therapy in the treatment of HNC. Frequently, the expression of *PD-L1* is high in HNC neoplasms. CheckMate141 assessed the effect of nivolumab compared to an investigator’s choice (IC) in squamous cell carcinoma of HNC. A total of 240 patients received nivolumab (3 mg/kg, every two weeks) while 121 patients received an IC. The OS was the primary endpoint. Nivolumab improved OS vs. IC (16.9% vs. 6.0%, respectively). Grade 3–4 irAEs were 15.3% and 36.9% for nivolumab and IC, respectively [109]. Another phase II trial (KEYNOTE 040) has independently shown that OS in recurrent and/or metastatic HNSC patients, who have failed platinum-based therapy, can be increased with anti-PD-1 monotherapy (nivolumab or pembrolizumab). Nivolumab improved OS with a hazard ratio (HR) of 0.70 as compared to standard of care (SOC) chemotherapy regimens (cetuximab, docetaxel, or methotrexate). The OS in the chemotherapy group was different from CheckMate141 (6.9 months median in KEYNOTE 040 versus 5.1 months in CheckMate141) [110].

The combination of nivolumab and ipilimumab in HNC is being evaluated in several active ongoing trials (Table 8). As a result, immunotherapy options, especially ICIs, seem to have fundamentally substituted the treatment of HNC with nivolumab and other monoclonal antibodies such as pembrolizumab, already suggesting new curative treatment options in advanced and metastatic disease (74).

### 7.10. Combination Therapy in Urothelial Carcinoma

Urothelial carcinoma (UC), including urinary bladder cancer (UBC) and upper tract urothelial carcinoma (UTUC), is one of the most common cancers in the USA [111]. Nearly all cases of UC are UBC, while UTUC is considered in just 5–10% of all UCs [112]. Platinum-based chemotherapy is the standard therapy for patients. Recently, the advent of ICIs in the treatment of UBC leads to the improvement of treatment options for patients in advanced stages [113]. Among ICIs, five monoclonal antibodies including atezolizumab, pembrolizumab, avelumab, durvalumab, and nivolumab have previously received FDA approval in UCs [114]. In a multi-center, phase 2 trial, Sharma P et al. assessed the safety and activity of nivolumab (3 mg/kg) in metastatic UC. A total of 265 patients in 11 countries were treated with nivolumab (3 mg/kg). The primary endpoint was ORR. The median OS was 7 months and confirmed ORR was achieved in 52 (19.6%) of patients. Also, in 81 patients who had PD-L1 expression of 5% or greater, ORR was obtained in 23 (28.4%) of them [115]. CheckMate 032 evaluated the efficacy of nivolumab in recurrent/advanced UC. Patients received nivolumab (3 mg/kg IV) every two weeks until disease progression. The OR was obtained in 19 of 78 patients. Serious side effects were reported in 36 (46%) of 78 patients, and nivolumab monotherapy showed a durable clinical response and a safety profile [116]. CheckMate 275 evaluated the efficacy of nivolumab in patients with advanced or metastatic UC. A total of 270 patients were treated with nivolumab. The ORR was 20.4%; mPFS was 1.9 (95%CI: 1.9–2.3); and the median OS was 8.6. The most common irAEs were fatigue (18.1%) and diarrhea (12.2%) [117].

On the whole, ICIs have shown beneficial clinical activity in patients with advanced UC as first- and second-line therapy. Ongoing investigations will help determine the optimal dosage, possible adverse effects, and combination approaches.

### 7.11. Resistance to Immune Checkpoint Therapy

Although ICIs offer durable clinical responses, more than 80% of the patients fail to respond to ICIs, thus showing innate and acquired resistance and ultimately experience a relapse of malignancy [118]. The endurance of the responses to immune checkpoints can be considered as a survival benefit and is one of the hallmarks of immunotherapy. However, in approved agents, only a group of tumor histologies and a small percentage of the patients in each histology are responding to these agents. There are various and multifactorial processes leading to both primary and acquired resistance to blocked checkpoints and they are based on individual environmental and genetic factors [119]. Several studies have examined primary resistance as an event in which a patient’s CD8+ T cells are either incapable of recognizing and localizing the tumor or are ineffective despite adequate localization. The second mechanism can occur due to other cells that exert local immunosuppressive outcomes within the tumor microenvironment [25]. Insufficient antitumor T cell generation [120], inadequate antitumor T cell effector function [121], and the impaired formation of T cell memory are important resistance mechanisms, both primary and acquired, to immune checkpoints (Table 9).

### 7.12. Immune-Related Adverse Events of Nivolumab and Ipilimumab

Enhancing immune responses using ICIs induces activation of T cell responses systemically, producing a variety of adverse effects and autoimmune toxicities called immune-related adverse events (irAEs). IrAEs commonly occur within the first six months of therapy. Although the frequency of adverse events varies depending on the monoclonal antibodies used and the type of cancer, the form of these toxicities is the same [134]. Grade 3 and 4 (grade 3: severe adverse event, grade 4: life-threatening or disabling adverse events) irAEs are reported in more than 55% of patients treated with the combination of ipilimumab 3 mg/kg and nivolumab 1 mg/kg. Adverse events are generally manageable with established guidelines, including the use of corticosteroids for grade 3 or 4 events [70]. Based on the evidence, up to 90% of patients treated with an anti-CTLA-4 antibody and 70% of patients treated with a PD-1/PD-L1 antibody develop irAEs, which affect any of the body organs including thyroiditis, dermatitis, pneumonitis, colitis, hepatitis, hypophysitis, uveitis, polyneuritis, and pancreatitis [135]. Cutaneous irAEs are the most common with anti-CTLA4/anti-PD1 monotherapy and the combination of these inhibitors. Rash and erythema are the most common cutaneous irAE (40%) for patients who are treated with combination therapy compared to 25% in anti-CTLA-4, and 15% in anti-PD1 monotherapies [136]. Pruritus is another common cutaneous irAE in patients who received the combination (33%) compared to 25–35% in patients treated with ipilimumab and 13.2% with nivolumab [137]. For grade 1 and 2 cutaneous irAEs, researchers suggested continuing with treatment and using topical steroids and topical emollients. For higher grades (3 and 4), treatment should be discontinued, and corticosteroids should be considered as the treatment of irAEs. The irAEs in the gastrointestinal (GI) tract is higher with anti-CTLA4 than with anti-PD1 therapy and increases with a combination of both and may result in diarrhea, colitis, or hepatitis [138]. Diarrhea was reported in 33/6%, 19/2%, 44/9% with anti-CTLA4, anti-PD1, and combination therapy, respectively [39]. Grade 1 and 2 diarrhea may be controlled with antidiarrheals and treatment can be continued. For higher grades, intravenous corticosteroids (methylprednisolone 1–2mg/Kg per day) for several weeks are the first line of therapy until stabilization [139]. In addition, hepatitis is higher in patients who were treated with combination therapy (14–18%) than patients who were treated with anti-CTLA4 (4–9%) or anti-PD1 (1–4%) monotherapy [38]. In grade 1 hepatitis, treatment may be continued with liver test monitoring. In grade 2 hepatitis, treatment should be delayed and corticosteroids (1 mg/kg/day) should be considered in case of persistent anomalies [140].

Endocrine irAEs, including hypothyroidism, hyperthyroidism, and hypophysitis, are common in around 30% of patients receiving nivolumab and ipilimumab combination therapy versus 14.4% and 10.9% with nivolumab and ipilimumab, respectively [141,142]. Hypophysitis is another common endocrine observed irAEs after combination therapy and can affect around 8% of patients. Hypophysitis leads to a low release of pituitary gland hormones, including prolactin, ACTH, TSH, FSH, luteinizing hormone (LH), and growth hormone [143]. Interstitial lung irAEs, such as pneumonitis, happen in approximately 10% of patients taking combination therapy compared to3% in PD-1/CTLA-4 monotherapy, and therefore, need particular attention from a physician to examine respiratory symptoms [144]. The therapeutic management of patients with grade 1 to 2 interstitial lung irAEs is with oral corticosteroids at 1 mg/kg daily and treatment delay is not effective alone regarding the long half-life of ICIs [145]. Eye irAEs have been seen in patients taking ICIs. These adverse effects include inflammation, uveitis, conjunctivitis, and episcleritis [146]. The incidence of neurological irAEs is around 12% with the combination compared to 4% with anti-CTLA4 and 6% for anti-PD1 antibodies. The incidence of kidney disease is rare and has been reported in 1% of patients [147]. Common renal diseases, including nephritis, granulomatous nephritis, renal enlargement due to inflammation have been noticed [148]. In patients receiving ipilimumab, hematologic syndromes such as cytopenia are rarely reported in solid tumors and are more common in lymphoma patients [149] (Table 10).

Collectively, clinical trials reported increased toxicity with the combination of ICIs. Altering schedules and/or dosing of ipilimumab to try to minimize irAEs while maintaining efficacy is one of the possible approaches [150]. It seems lowering the administration dose of ipilimumab to 1 mg/kg in combination with nivolumab or administering ipilimumab less frequently (i.e., every 6 or 12 weeks) has been proper approaches to try to minimize irAEs [151].

Corticosteroids are the main agents for low-severity irAEs (grades 1–2), administered at low (0.5–1 mg/kg/day), moderate (1–2 mg/kg/day), or high dosages (>2 mg/kg/day). After the resolution of irAEs, patients will require progressively decreasing corticosteroid therapy. Other immunosuppressive drugs may be admitted if there are severe irAEs (grades 3–4) or when the use of corticosteroids does not resolve irAEs. On the whole, ICIs therapy may be recommenced while most grade 1 events are managed. For grades 2–4, immunotherapy is usually delayed and can be reinitiated once irAEs are resolved, although permanent discontinuation is sometimes required [152].

## 8. Expert Commentary

The combination of nivolumab and ipilimumab in different cancers led to higher overall survival and objective response rates than chemotherapy or nivolumab/ipilimumab monotherapy [155]. Although the combination of nivolumab and ipilimumab therapy has been a great advancement in cancer treatment, several challenges such as immune-associated toxicity, treatment resistance, and patient selection remain unresolved. The higher benefit of combination therapy comes at the cost of higher irAEs, demanding proper patient selection, and counseling. Patient selection is one of the most important considerations for future research on the field of combination immunotherapy [156]. Patients should be capable of handling the possibility of irAEs and following treatment with corticosteroids. Since the management of irAEs and application of immunosuppressive treatment regimens require close communication, patients who have difficulty in communicating with healthcare workers may not be good candidates for combination immunotherapy. The question of whether ipilimumab and nivolumab combination is superior to nivolumab alone remains another dilemma for clinicians who must decide between these treatments for their patients. Clinical responses to ICIs are variable. The identification of biomarkers to predict response and treatment-mediated toxicity remains another important unresolved dilemma. Several biomarkers have been found promising. For example, PD-L1 expression, high mutational load, selective CD8+ T cell infiltration, and distribution at tumor invasive margins correlate with clinical response to anti-PD-1/PD-L1treatment [11]. The identification of these biomarkers helps physicians to choose a rational treatment for patients. In addition, several clinical trials combining ICIs either with chemotherapy, targeted therapy, or other immunotherapies reach better outcomes with an acceptable safety profile. There is no doubt that many similar combinations will be reported in the near future. ICIs combination will likely be the next paradigm of care for various cancers. This declaration can only be achievable with better perception, prevention, and management of irAEs associated with these combinations.

## 9. Conclusions

The recent advances in immunotherapy and the approval of ICIs such as ipilimumab and nivolumab have revolutionized the treatment of cancer. However, all patients cannot benefit from checkpoint inhibitors monotherapy. To overcome this, a combination of these agents is frequently being investigated as an approach to improve outcomes. Despite their efficacy, these agents also cause immune-related adverse effects that may be life-threatening if not detected and controlled appropriately. Overall, oncologists should cooperate with clinical immunologists to understand, control, and limit the toxicity of immunotherapy.

## Figures and Tables

**Figure 1 ijms-21-04427-f001:**
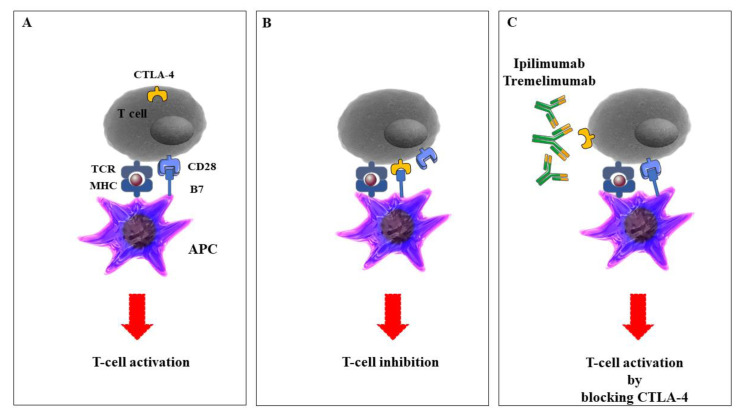
The role of cytotoxic T-lymphocyte-associated protein 4 (CTLA-4) inhibitors in the activation of T cells. A: Antigen-presenting cells (APCs), including dendritic cells (DCs), macrophages, natural killer (NK) cells, and B cells, process tumor antigens and present them to specific T cells, leading to activation of the T cells and immune responses to the tumor. B: Upon T cell receptor activation, CTLA-4 is expressed on the T cell surface and interacts with the co-receptor CD28 that is expressed on APCs, leading to the end of the T cell responses. C: Anti-CTLA-4—specific monoclonal antibodies prevent the interaction between CTLA-4 and CD28 and contribute to inhibitory signals in T cells. The figure was produced using Servier Medical Art (http://smart.servier.com/).

**Figure 2 ijms-21-04427-f002:**
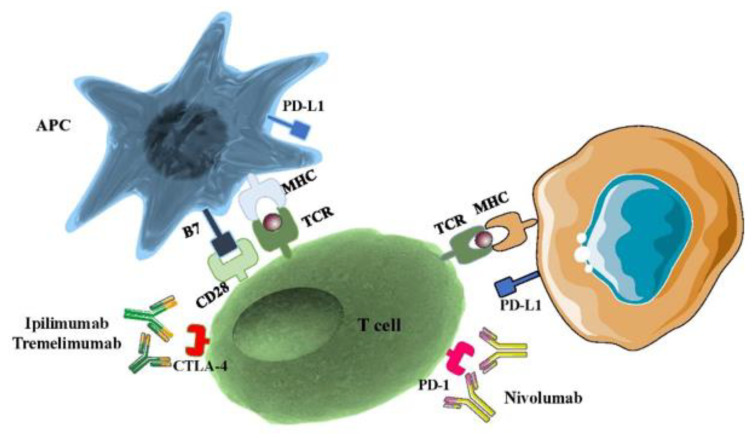
Mechanism of CTLA-4 and PD-1 inhibition.

**Table 1 ijms-21-04427-t001:** Summary of completed (until January 2020) clinical trials of ipilimumab plus nivolumab in unresectable or metastatic melanoma.

Reference	Trial Phase	Treatment Arms	Primary Endpoints	Results
**[61]**	2	Induction Phase: Nivolumab + Ipilimumab infusion (IV) Maintenance Phase: Nivolumab infusion (IV)	Intracranial CBR (up to six months)	The rate of intracranial CBR was 57% The rate of CR was 26%, the rate of PR was 30%, and the rate of SD for at least 6 months was 2%.
**[62]**	3	Arm A: Nivolumab+ Placebo for Ipilimumab+ Placebo for Nivolumab Arm B: Nivolumab+ Ipilimumab+ Placebo for Nivolumab	Rate of PFS Rate of OS (Time Frame: 6, 12, and 24 months)	The OS rate at 3 years was 58% in the nivolumab-plus-ipilimumab group and 52% in the nivolumab group, as compared with 34% in the ipilimumab group.
**[63]**	2	Cohort 1 Nivolumab Monotherapy (nivolumab 3 mg/kg every 2 weeks) Cohort 2 Nivolumab and Ipilimumab (nivolumab 1 mg/kg combined with ipilimumab 3 mg/kg every 3 weeks for four doses)	Intracranial response rate (at 3 years)	Intracranial responses were achieved by 20% of patients in cohort 1 and 46% of patients in cohort 2. Intracranial complete responses occurred in 12% of patients in cohort 1 and 17% of patients in cohort 2.
**[64]**	3	Arm A: Nivolumab+ Placebo for Ipilimumab+ Placebo for Nivolumab Arm B: Nivolumab+ Ipilimumab+ Placebo for Nivolumab	Rate of PFS Rate of OS(Time Frame: 6, 12, and 24 months)	The median OS was more than 60.0 months in the nivolumab-plus-ipilimumab group and 36.9 months in the nivolumab group, as compared with 19.9 months in the ipilimumab group. The OS at 5 years was 52% in the nivolumab-plus-ipilimumab group and 44% in the nivolumab group, as compared with 26% in the ipilimumab group.
**[65]**	2	Arm 1: Nivolumab (1 mg/kg+ Ipilimumab (3 mg/kg) Arm 2: Placebo + Ipilimumab	Percentage of participants with OR in the randomized, BRAF wild-type population (at a minimum of 6 months)	Among patients with BRAF wild-type tumors, the rate of OR was 61% in the combination group versus 11% in the ipilimumab-monotherapy group), with CR reported in 22% in the combination group and no patients in the ipilimumab-monotherapy group.

**Table 2 ijms-21-04427-t002:** (Until January 2020) clinical trials of ipilimumab plus nivolumab in RCC.

End Time	Trial Phase	Enrollment	Primary Endpoints	Treatment Arms	Clinical Trials Identifier
2037	2	74 patients	ORR at two years	Nivolumab (240 mg every 2 weeks during the first 20 weeks, 480 mg every 4 weeks thereafter and Ipilimumab (After 2 weeks 1mg/kg every 6 weeks)	NCT03297593
2021	2	120 patients	PFS rate at one year	Arm A: Nivolumab (240mg and 360mg) Arm B: Nivolumab (3mg/kg) + Ipilimumab (1mg/kg)	NCT03117309
2021	2	53patients	Establish the recommended Phase II dose (RP2D) at 6 months ORR at two years	Entinostat (5mg, 3mg, or 2mg orally (PO) on D1, 8, 15), Nivolumab (3 mg/kg IV D1 and Ipilimumab 1 mg/kg IV D1)	NCT03552380
2024	3	676patients	Duration of PFS (Time Frame: up to 23 months)	Experimental Arm: Cabozantinib + nivolumab + ipilimumab (4 doses) followed by cabozantinib + nivolumab Control Arm: Cabozantinib-matched placebo + nivolumab + ipilimumab (4 doses) followed by cabozantinib-matched placebo + nivolumab	NCT03937219

**Table 3 ijms-21-04427-t003:** Ongoing trials of ipilimumab plus nivolumab in Metastatic and Microsatellite Stable CRC.

End Time	Enrollment	Trial Phase	Primary Endpoints	Treatment Arms	Clinical Trials Identifier
2021	32	1	To determine the recommended dose level of the combination of regorafenib, nivolumab, and ipilimumab in patients with advanced metastatic RCC	Patients receive regorafenib on days 1–21, nivolumab, and ipilimumab IV. Cycles repeat every 28 days for up to 2 years	NCT04362839
2022	100	2	The 8-month PFS rate	Temozolomide 150 mg/sqm daily on days 1–5 every 4 weeks, for two cycles followed by TC scan assessment, nivolumab 480 mg i.v. every 4 weeks, low-dose ipilimumab 1 mg/Kg i.v. every 8 weeks and temozolomide	NCT03832621
2025	494	3	PFS (Time Frame: Up to 5 years)	Arm A: Nivolumab Monotherapy Arm B: Nivolumab + Ipilimumab Combination Arm C: Investigator’s Choice Chemotherapy	NCT04008030
2024	80	2	Disease control rate (Time Frame: 2 years)	Nivolumab (3 times per cycle) +Ipilimumab (once per cycle) Radiation Therapy	NCT03104439

**Table 4 ijms-21-04427-t004:** Ongoing (until January 2020) clinical trials ipilimumab plus nivolumab in SCLC.

Estimated time	Enrollment	Trial Phase	Primary Endpoints	Treatment Arms	Clinical Trial Identifier
2017–2022	21 participants	Phase 1/2	PFS (Time Frame: 6 months)	Thoracic Radiation Therapy (3Gy × 10 fractions) for 10 days Ipilimumab 3 mg/kg (90 min IV infusion) every 3 weeks plus Nivolumab 1 mg/kg (30 min IV infusion) will be administered every 3 weeks	NCT03043599
2018–2021	41 participants	Phase 2	Disease Control Rate (DCR) (TimeFrame: up to 3 years)	Combination immunotherapy with Ipilimumab and Nivolumab plus a Dendritic Cell-based p53 Vaccine (Ad.p53-DC)	NCT03406715
2014–2022	264 participants	Phase 2	The OS and PFS rates (at a maximum of 6,5 years)	Induction: Nivolumab at a dose of 1 mg/kg i.v. followed (on the same day) by Ipilimumab at a dose of 3 mg/kg i.v. once every 3 weeks, 4 cycles Maintenance: Nivolumab 240 mg i.v. once every 2 weeks, for a maximum of 12 months from the start of maintenance	NCT02046733
2018–2022	55 participants	Phase 1/2	Phase I: Maximum tolerated dose (MTD) (Time Frame: 9 Months) Phase II: PFS (Time Frame: 36 Months)	Phase I: nivolumab, ipilimumab, and plinabulin Phase II Arm A: nivolumab and ipilimumab Phase II Arm B: nivolumab, ipilimumab, and plinabulin	NCT03575793

**Table 5 ijms-21-04427-t005:** Ongoing (until January 2020) clinical trials of ipilimumab plus nivolumab in NSCLC.

End Time	Enrollment	Trial Phase	Primary Endpoints	Treatment Arms	Clinical Trial Identifier
2020	184	2	The ORR at two years	Arm 1: Nivolumab (3 mg/kg, every two weeks) Arm 2: Nivolumab (3 mg/kg, every two weeks) and Ipilimumab (1 mg/kg, every six weeks)	NCT03091491
2020	472	1	Number of participants who experienced serious adverse events and adverse events, the number of participants who experienced selected adverse Events, and the number of participants with abnormalities in selected hepatic and thyroid clinical laboratory tests	Nivolumab in combination with Gemcitabine, Cisplatin, Pemetrexed, Paclitaxel, Carboplatin, Bevacizumab, Erlotinib, and Ipilimumab in different arms	NCT01454102
2025	580	3	PFS (Time Frame: up to 47 months)	Arm 1: Nivolumab+Platinum doublet chemotherapy Arm 2: Nivolumab + Ipilimumab Arm 3: Platinum doublet chemotherapy	NCT02864251

**Table 6 ijms-21-04427-t006:** Ongoing (until January 2020) esophageal cancer trials evaluating Nivolumab/Ipilimumab combination.

End Time	Enrollment	Phase	Primary Endpoints	Treatment Arms	Clinical Trial Identifier
**2022**	130	2	12-months PFS	Arm A: Chemoradiation (50Gy in 25 fractions over 5 weeks (i.e., 2Gy per fraction), concurrently with 3 cycles of 2 weeks of FOLFOX) + Nivolumab (IV 240 mg on days 1, 15 and 29) Arm B: Chemoradiation + Nivolumab + Ipilimumab (IV 1 mg/kg on day 1 followed by a maintenance phase)	NCT03437200
**2021**	939	3	OS and PFS	Arm A: Nivolumab + Ipilimumab Arm B: Nivolumab + Cisplatin + Fluorouacil Arm C: Cisplatin + Fluorouracil	NCT03143153
**2021**	75	2	OS (Time Frame:36 months)	Arm A: Nivolumab/Ipilimumab combination treatment B. Nivolumab monotherapy	NCT03416244
**2023**	278	2/3	Pathologic CR (Step I) (Time Frame: Up to 5 weeks) Disease-free survival (DFS) (Step 2)	Arm A (carboplatin, paclitaxel, radiation therapy) Arm B (carboplatin, paclitaxel, radiation therapy, nivolumab) Arm C (nivolumab) Arm D (nivolumab, ipilimumab)	NCT03604991
**2022**	97	2	OS (at 12 months)	Arm A: Chemo-free immunotherapy with Nivolumab, Ipilimumab, Trastuzumab Arm B: Addition of Nivolumab to Standard therapy (chemotherapy and Trastuzumab)	NCT03409848

**Table 7 ijms-21-04427-t007:** Ongoing (until January 2020) Advanced Hepatocellular Carcinoma trials assessing Nivolumab/Ipilimumab combination.

End Time	Enrollment	Trial Phase	Primary Endpoints	Treatment Arms	Clinical Trial Identifier
2022	32	1/2	Delay to surgery (Time Frame: Up to Day 89) Incidence of treatment-emergent adverse events (Time Frame: Up to Day 127)	Ipilimumab (1 mg/kg, once every 3 weeks, for 3 weeks) + Nivolumab (3 mg/kg, once every 3 weeks, for 6 weeks)	NCT03682276
2023	1084	3	OS (Time Frame: up to 4 years)	Arm A: Nivolumab + Ipilimumab Arm B: Sorafenib/Lenvatinib	NCT04039607
2024	12	1	Drug-related toxicities (Time Frame: 4 years) Fold change in interferon-producing DNAJB1-PRKACA-specific cluster of differentiation 8(CD8) and 4 (CF4) T cells	DNAJB1-PRKACA peptide vaccine, Nivolumab, and Ipilimumab	NCT04248569
2022	1097	1/2	Safety and Tolerability of nivolumab ORR, Safety, and Tolerability of nivolumab plus ipilimumab	Non-infected: Nivolumab HCV-infected: Nivolumab HBV-infected: Nivolumab Nivolumab plus Ipilimumab Combination Nivolumab plus Cabozantinib Combination Nivolumab plus Ipilimumab plus Cabozantinib	NCT01658878
2022	32	1/2		Ipilimumab Nivolumab	NCT03682276

**Table 8 ijms-21-04427-t008:** Ongoing (until January 2020) Head and Neck Cancer trials evaluating nivolumab plus ipilimumab.

End Time	Enrollment	Trial Phase	Primary Endpoints	Treatment Arms	Clinical Trial Identifier
2022	24	1	Incidence of adverse events (Time Frame: Up to 6 months)	Nivolumab Ipilimumab Radiation Therapy	NCT03162731
2024	60	2	Adverse Events related to treatment (Time Frame: Up to 4 months)	Arm A: Nivolumab + Relatlimab Arm B: Nivolumab + Ipilimumab Arm C: Nivolumab	NCT04080804
2024	140	2	2 years of disease-free survival	Arm A: Nivolumab Arm B: Ipilimumab+ Nivolumab	NCT03406247
2024	40	2	Response rates to treatment (Time Frame: at time of surgery)	Arm A: Nivolumab Arm B: Ipilimumab+ Nivolumab	NCT02919683
2020	36	1	Change in immune profile in the tumor microenvironment Change in circulating percentage of immune suppressor subsets in peripheral blood Phenotypic shifts in T cell subsets in peripheral blood	Group A (VX15/2503) Group B (VX15/2503, ipilimumab) Group C (VX15/2503, nivolumab) Group D (nivolumab) Group E (ipilimumab)	NCT03690986
2026	947	3	OS in participants with PD-L1 expressing tumors. (Time Frame: Approximately 51 months) OS in all participants	Experimental: Nivolumab and Ipilimumab Active Comparator: Extreme Regimen	NCT02741570
2024	675	2	ORR in the platinum-refractory subgroup (Time Frame: 28 months) Duration of response in the platinum-refractory subgroup (Time Frame: 28 months)	Experimental: Nivolumab and Ipilimumab Active Comparator: Nivolumab and Ipilimumab-placebo	NCT02823574
2024	276	3	Disease-free survival (Time Frame: approximately 71 months)	Experimental: Neoadjuvant/adjuvant Nivolumab and Ipilimumab Active Comparator: Surgical resection + adjuvant radio(-chemo) therapy	NCT03700905
2021	32	1/2	The number of patients that will not endure a delay in surgery Tumor response to NAI The potential impact of local tumor hypoxia on tumor T cell abundance (Time Frame: 2.5 years)	Experimental: Nivolumab with or without Ipilimumab	NCT03003637

**Table 9 ijms-21-04427-t009:** Mechanism of primary and secondary resistance to checkpoint blocked.

Primary Resistance	Reference	Secondary Resistance	Reference
Presence of inactivating mutations in JAK1, JAK2, and beta2-microglobulin (B2M)	[122]	Inactivating mutations in beta2-microglobulin (B2M)	[123]
Lower MHC-I expression	[124]	Increased PD-L2 expression on PD-L1 negative tumor cells	[125]
Overexpression of VEGF	[126]	PD-L1 up-regulation	[127]
Activation of PI3K/AKT, ALK/STAT3, and MEK/ERK/STAT1 signaling pathways	[128]	JAK1/2 mutation	[129]
TGF-β signaling pathway	[130]		
Epithelial-mesenchymal transition (EMT)	[131]		
Exhaustion of T cells	[132]		
Increase in Tumor-associated macrophage and Myeloid-derived suppressor cells (MDSCs)	[133]		

**Table 10 ijms-21-04427-t010:** The incidence of irAEs induced by immune checkpoint inhibitors [153,154].

Common irAEs	CTLA-4 Inhibitors	PD-1 Inhibitors	Combination of Nivolumab and Ipilimumab
Cutaneous			
Rash	34%	10–21%	30%
Pruritus	25–30%	11–21%	35%
Vitiligo	4%	11%	9%
Gastrointestinal Disease			
Diarrhea	38%	8–20%	45%
Colitis	8–10%	1–3%	13%
Neurological Disease	4%	6%	12%
Endocrine system			
Hypothyroidism	1–2%	4–10%	17%
Hyperthyroidism	2–3%	Less than 1%	7%
Lung			
Pneumonitis	Less than 1%	1–5%	7%
Liver			
Hepatitis	Less than 1%	1–2%	14–18%

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
