# Peer review of "Combination of Ipilimumab and Nivolumab in Cancers: From Clinical Practice to Ongoing Clinical Trials"

_ijms, 2020, doi:10.3390/ijms21124427_

Round 1

Reviewer 1 Report

The authors provide a comprehensive review of Ipilimumab and Nivolumab in various cancers. The manuscript is comprehensive in terms of the amount of information.

Following points need to be addressed:

  1. What is the novelty of the current review? How does this provide information that would advance the knowledge in the field? It has to be mentioned in the abstract and the introduction section
  2. Since most of the tables provide completed or in progress clinical trials, I strongly suggest the authors to include the primary endpoints type of clinical trial, and key results/conclusion in the table for the completed studies.
  3. In addition, it is essential to mention the search criteria, inclusion/exclusion criteria of studies by the authors before selecting to include a study in the table. Most of the studies are directly searched from one source ClinicalTrials.gov website and thus it is essential to provide the reason for it.
  4. Finally since this is a review, I suggest the authors to include a final expert opinion paragraph as to the current state of the art and the future in terms of types of clinical studies that should be performed.

Author Response

Reviewer 1:

  1. What is the novelty of the current review? How does this provide information that would advance the knowledge in the field? It has to be mentioned in the abstract and the introduction section.

We appreciate your comment. Clinical trials showed a significant increase of the response rates in patients who received a combination of ipilimumab and nivolumab compared to ipilimumab/nivolumab monotherapy and this combination received FDA approval as first-line treatment for patients with metastatic melanoma, advanced renal cell carcinoma, metastatic colorectal cancer with MMR/MSI-H aberrations, and metastatic or recurrent non-small cell lung cancer. Despite promising results for the combination of ipilimumab and nivolumab, safety concerns slowed down the development of such strategies. So, we evaluated data concerning the clinical activity and the adverse events of ipilimumab and nivolumab combination therapy, assessing ongoing clinical trials to identify clinical outlines that may support combination therapy as an effective treatment. To the best of our knowledge, this paper is one of the first studies to evaluate the efficacy and safety of ipilimumab and nivolumab combination therapy in several cancers.

  1. Since most of the tables provide completed or in progress clinical trials, I strongly suggest the authors to include the primary endpoints type of clinical trial, and key results/conclusion in the table for the completed studies.

Thank you for your suggestion. Tables thoroughly revised and the primary endpoints with treatment arms added to the newer version. Only Table 1 presents completed clinical trials and the results of these studies added to Table 1. Other tables cover ongoing trials that are under investigation and for now, there is no available result for them.

  1. In addition, it is essential to mention the search criteria, inclusion/exclusion criteria of studies by the authors before selecting to include a study in the table. Most of the studies are directly searched from one source ClinicalTrials.gov website and thus it is essential to provide the reason for it.

Thank you for your suggestion. The new section entitled “Methodology” added to the new version of manuscript.Since ClinicalTrials.gov is the largest and reliable trial registry and in the world, we searched Clinical-Trials.gov for ongoing and completed clinical trials until January 2020.

  1. Finally since this is a review, I suggest the authors to include a final expert opinion paragraph as to the current state of the art and the future in terms of types of clinical studies that should be performed.

Thank you for your suggestion. The new section entitled “Expert commentary” added to the revised manuscript.

Reviewer 2 Report

Authors summarized the current evidence about nivolumab and ipilimumab. This combination therapy has been expected improve prognosis in various types of malignant tumors. I have some comments as follows.

  1. Tables do not include important information such as primary endpoint, treatment arm, treatment line (1st, 2nd or salvage line). These information would be very important to see clinical trials.
  2. Regarding pharmacology, authors should describe PK/PD data of these drug. Since there are some the dosage and treatment schedule of ipilimumab, it would be better to describe how to be decided the dose of ipilimumab.
  3. Authors introduced many ongoing clinical trials. However, my concern about this combination therapy is a management of immune-related adverse event. Please describe more about irAE. What is the difference of monotherapy and how to manage them.

Author Response

  1. Tables do not include important information such as primary endpoint, treatment arm, treatment line (1st, 2nd or salvage line). These information would be very important to see clinical trials.

Thank you for your suggestion. Tables thoroughly revised and the primary endpoints with treatment arms added to the current version of manuscript.

  1. Regarding pharmacology, authors should describe PK/PD data of these drug. Since there are some the dosage and treatment schedule of ipilimumab, it would be better to describe how to be decided the dose of ipilimumab.

Thank you for your suggestion. The treatment schedule and dosage of ipilimumab added to the section of “Ipilimumab pharmacology” and related cancers, highlighted with yellow.

  1. Authors introduced many ongoing clinical trials. However, my concern about this combination therapy is a management of immune-related adverse event. Please describe more about irAE. What is the difference of monotherapy and how to manage them?

Thank you for your suggestion. In the newer version of the manuscript, we described more about combination therapy and irAEs. We provided a new table that compared the incidence of different irAEs in anti-CTLA-4/anti-PD-1 monotherapies and combination therapy. Also, more information about how we can manage irAEs during immune checkpoint therapy added to the new manuscript.

Round 2

Reviewer 1 Report

The authors have addressed the comments that were raised in my previous review and so I recommend the manuscript for acceptance to the editor

Reviewer 2 Report

Tables are improved well. Also, the discussion about irAE is also well described.